# A Proposed Algorithm for Endoscopic Ultrasound-Guided Rendezvous Technique in Failed Biliary Cannulation

**DOI:** 10.3390/jcm9123879

**Published:** 2020-11-29

**Authors:** Saburo Matsubara, Keito Nakagawa, Kentaro Suda, Takeshi Otsuka, Hiroyuki Isayama, Yousuke Nakai, Masashi Oka, Sumiko Nagoshi

**Affiliations:** 1Saitama Medical Center, Department of Gastroenterology and Hepatology, Saitama Medical University, Saitama 350-8550, Japan; kate-ill@hotmail.co.jp (K.N.); leclearlshelly@gmail.com (K.S.); ohitoyosinokaze@yahoo.co.jp (T.O.); moka@saitama-med.ac.jp (M.O.); snagoshi@saitama-med.ac.jp (S.N.); 2Department of Gastroenterology, Graduate School of Medicine, Juntendo University, Tokyo 113-8421, Japan; h-isayama@juntendo.ac.jp; 3Department of Gastroenterology, Graduate School of Medicine, The University of Tokyo, Tokyo 113-8654, Japan; ynakai.tky@gmail.com

**Keywords:** endoscopic ultrasound, endoscopic retrograde cholangiopancreatography, rendezvous, biliary cannulation, algorithm

## Abstract

Background: The selection of an approach route in endoscopic ultrasound-guided rendezvous (EUS-RV) for failed biliary cannulation is complicated. We proposed an algorithm for EUS-RV. Methods: We retrospectively evaluated consecutive EUS-RV cases between April 2017 and July 2020. Puncturing the distal extrahepatic bile duct (EHBD) from the duodenal second part (D2) (DEHBD/D2 route) was attempted first. If necessary, puncturing the proximal EHBD from the duodenal bulb (D1) (PEHBD/D1 route), puncturing the left intrahepatic bile duct (IHBD) from the stomach (LIHBD/S route), or puncturing the right IHBD from the D1 (RIHBD/D1 route) were attempted in this order. Results: A total of 16 patients were included. The DEHBD/D2 route was used in 10 (62.5%) patients. The PEHBD/D1 route was attempted in five (31.3%) patients, and the biliary puncture failed in one patient in whom the RIHBD/D1 route was used because of tumor invasion to the left hepatic lobe. The LIHBD/S route was applied in one (6.3%) patient. Successful biliary cannulation was achieved in all patients eventually. The time from the puncture to the guidewire placement in the DEHBD/D2 route (3.5 min) was shorter than that in other methods (14.0 min) (*p* = 0.014). Adverse events occurred in one (6.3%) patient with moderate pancreatitis. Conclusions: The proposed algorithm might be useful for the selection of an appropriate approach route in EUS-RV.

## 1. Introduction

Endoscopic retrograde cholangiopancreatography (ERCP) is the standard procedure to relieve various biliary disorders. Selective deep biliary cannulation is an essential step of ERCP. However, failed cannulation can occur even after the application of a number of biliary cannulation techniques [1].

Endoscopic ultrasound-guided rendezvous (EUS-RV) is one of the rescue techniques for failed biliary access in therapeutic ERCP [2,3,4,5,6,7,8,9,10,11,12]. Several approach routes in EUS-RV have been reported with regard to the scope position and puncture site, such as the puncture of the distal extrahepatic bile duct (EHBD) from the duodenal second part (D2) with a stretched, short scope position (DEHBD/D2 route); the puncture of the proximal EHBD from the duodenal bulb (D1) with a pushed, long scope position (PEHBD/D1 route); and the puncture of the left intrahepatic bile duct (IHBD) from the stomach with a short scope position (LIHBD/S route) [2]. These approach routes have different properties in many respects: scope stability, needle handling, and the diameter of the bile duct, which are associated with the biliary puncture; the distance between the puncture site and the ampulla, the direction of the needle, and the passage or not through the stricture, which are associated with the guidewire manipulation. Alternately, scope stability, the diameter of the bile duct, and the passage or not through the stricture are affected by the kind of biliary disorder, the stricture site, or the location and size of tumors. Moreover, the size of a puncture needle has impacts on the feasibility of the biliary puncture and on the selection of a guidewire. These many complicated factors make the selection of an approach route difficult. Given that the change of approach routes after failed initial attempts is time-consuming and harmful and possibly hampers the procedure completion, a standard algorithm for the selection of an appropriate approach route is desirable. In this study, we aimed to estimate our experiences of EUS-RV performed in conformity with our method and to propose a new algorithm based on analyses of our experiences and a literature review.

## 2. Patients and Methods

### 2.1. Patients

This was a single center, retrospective cohort study using prospectively collected ERCP and interventional EUS data at Saitama Medical Center, Saitama Medical University. We extracted data on consecutive EUS-RV for failed biliary cannulation in therapeutic ERCP with a native papilla between April 2017 and July 2020. Patients with a surgically altered anatomy, except for Billroth 1 reconstruction, were excluded. The study was approved by the local ethical committee.

### 2.2. Our Cannulation Algorithm in Therapeutic Biliary ERCP

We usually performed wire-guided cannulation (WGC) in therapeutic biliary ERCP. The double guidewire technique (DGT) [13], followed by transpancreatic precut sphincterotomy (TPPS), was attempted for the difficult cannulation when the guidewire insertion to the pancreatic duct was achieved. If these advanced techniques or the pancreatic guidewire insertion failed, we performed EUS-guided biliary drainage (EUS-BD), including EUS-guided transmural biliary drainage and EUS-RV. In unresectable malignant cases, EUS-guided transmural drainage was preferably used because of its short procedural time and high success rate, while it was not suitable for benign cases or potentially resectable malignant cases due to a lack of evidence around the long-term outcomes or preoperative drainage [6]. Thus, we attempted EUS-RV in benign cases or potentially resectable malignant cases and in unresectable malignant cases unfit for EUS-guided transmural drainage.

### 2.3. EUS-RV Procedure

EUS-RV was performed by one endoscopist (S.M.) with enough experience of both EUS and ERCP related procedures, using an oblique-viewing curved-linear array echoendoscope (EG-580UT; Fujifilm Medical Corp, Tokyo, Japan or GF-UCT260; Olympus Medical Systems, Tokyo, Japan) and a dedicated processor (SU-1; Fujifilm Medical Corp, Tokyo, Japan or ME-2; Olympus Medical Systems, Tokyo, Japan). Patients were sedated with midazolam and pethidine hydrochloride while in the prone position. The bile duct was punctured using a 19 or 22 gauge fine-needle aspiration (FNA) needle (EZ Shot 3 Plus; Olympus Medical Systems, Tokyo, Japan) under ultrasound guidance, avoiding the intervening vessels evaluated on Doppler mode. After the confirmation of intraductal insertion by the aspiration of bile juice, a contrast agent diluted with saline was injected for the depiction of the biliary tree. Then, a 0.025 inch guidewire (EndoSelector; Boston Scientific Japan, Tokyo, Japan or VisiGlide2; Olympus Medical Systems, Tokyo, Japan) for a 19 gauge needle or a 0.018 inch guidewire (Pathfinder; Boston Scientific Japan, Tokyo, Japan) for a 22 gauge needle was inserted into the bile duct and advanced to the duodenum. When the guidewire manipulation was difficult, an ERCP catheter was inserted into the bile duct. Following the sequential withdrawal of the FNA needle and the echoendoscope leaving the guidewire in situ, a therapeutic duodenoscope (TJF-260V or JF-260V; Olympus Medical Systems, Tokyo, Japan) was inserted into the duodenum alongside the EUS-placed guidewire. After successful biliary cannulation was achieved, the EUS-placed guidewire was removed and planned therapeutic procedures were accomplished. Basically, biliary drainage tubes such as plastic stents, naso-biliary tubes, or self-expandable metal stents were placed across the puncture point to prevent the bile leak even without biliary strictures, especially when an ERCP catheter was inserted into the bile duct via the puncture point for the guidewire manipulation.

### 2.4. Our EUS-RV Method Regarding Approach Routes and Needle Sizes

There are three major steps in EUS-RV [14]: the biliary puncture, the guidewire manipulation, and the biliary cannulation. Of these, the guidewire manipulation is the most important issue due to technically difficulty [14] and a potential risk of guidewire shearing [15]. Moreover, failed cannulation after a successful biliary puncture may lead to a bile leak.

The guidewire manipulation is affected by several factors. First is the distance between the puncture site and the ampulla. A shorter distance makes the torqueability and pushability of the guidewire better and allows the guidewire to pass through the sphincter of Oddi more easily. Second is the direction of the needle. The guidewire can linearly proceed to the ampulla and easily pass through the sphincter of Oddi when the needle is directed toward the ampulla. Third is the positional relation between the puncture site and the stricture of the biliary tree in patients with biliary obstruction. The puncture at the proximal side of the stricture makes guidewire manipulation difficult due to two obstacles including the stricture and the sphincter of Oddi [14]. Fourth is the performance of the guidewire. Recently developed 0.025 inch guidewires have long hydrophilic tips with stiff shafts and facilitate guidewire manipulation and scope exchange, whereas 0.018 inch guidewires have poor torqueability and stiffness. For 22 gauge needles, only 0.018 inch guidewires can be used.

Given these factors, the (DEHBD/D2) route (Figure 1) using a 19 gauge needle with a 0.025 inch guidewire was primarily applied regardless of the stricture site because the distance between the puncture site and the ampulla and the direction of the needle were favorable for the guidewire manipulation. If the bile duct diameter was too small to be punctured with a 19 gauge needle, a 22 gauge needle could be used because the guidewire manipulation in this route was not so difficult, even with a 0.018 inch guidewire. On the other hand, a 22 gauge needle was not used in the other routes because of the difficulty of guidewire manipulation with a 0.018 inch guidewire. If the DEHBD/D2 route was impossible or unsuccessful, the PEHBD/D1 route (Figure 2) was applied due to the shorter distance between the puncture site and the ampulla and the higher possibility of avoiding the guidewire passage through the stricture owing to puncturing the distal side of the stricture than those of IHBD approaches. If necessary, the LIHBD/S route (Figure 3) or the puncture of the right IHBD from the D1 (RIHBD/D1 route) (Figure 4) were applied. The LIHBD/S route is preferable to the RIHBD/D1 route because of the flexibility of puncture point selection. A comparison of the four approach routes was noted in Table 1.

### 2.5. Biliary Cannulation Methods during EUS-RV

We used three cannulation methods including over-the-wire, along-the-wire, and hitch-and-ride [16]. In the over-the-wire method, an EUS-placed guidewire was caught by a snare and pulled through a channel of a duodenoscope, followed by the insertion of a catheter over the guidewire. In the along-the-wire method, standard WGC was applied along the EUS-placed guidewire. There are potential disadvantages in both methods: the risk of guidewire loss and time consumption in the former, and the unreliability of the biliary cannulation in the latter. To solve these problems, we used the hitch-and-ride method from April 2019, using a hand-made catheter that had a slit on the tip (Figure 5) to catch the EUS-placed guidewire during the along-the-wire method. We have reported the procedural details in a previous article [16].

### 2.6. Outcome Measures

We evaluated the rates of technical success and the adverse events of EUS-RV. The adverse events were graded according to the lexicon by the American Society for Gastrointestinal Endoscopy [17]. The reasons for the selection of the approach route and needle size regarding biliary disorders, the rates of and reasons for the conversion to other approach routes, and the outcome details of each approach route were also evaluated. Moreover, comparisons of the success rates and cannulation times between the three cannulation methods were performed.

### 2.7. Statistical Analyses

Descriptive continuous variables were presented as number (percentage) or median (range). Statistical comparisons were performed with Fisher’s exact test for discrete variables and with the Mann-Whitney U test or the Kruskal-Wallis test for continuous variables. *p*-values of 0.05 or less were considered statistically significant. All statistical analyses were performed with EZR Version 1.52 (Saitama Medical Center, Jichi Medical University, Saitama, Japan) [18], which is a graphical user interface for R (The R Foundation for Statistical Computing, Vienna, Austria).

## 3. Results

### 3.1. Baseline Characteristics

A total of 1759 ERCP procedures were performed during the study period. After excluding cases with prior ERCP (*n* = 756), pancreatic intervention (*n* = 129), surgically altered anatomy, except for Billroth 1 gastrectomy (*n* = 88), or inaccessible papillae (*n* = 40), 746 patients with native papillae for biliary intervention remained. Among them, biliary cannulation failed in 34 patients, even after using alternative techniques, including DGT or TPPS. Of these, EUS-RV was attempted in 16 patients with benign, potentially resectable malignant, or unresectable malignant disorders unfit for EUS-guided transmural drainage (Figure 6).

Indications of EUS-RV included bile duct stones in five, distal biliary obstruction in nine (three pancreatic carcinoma, three ampullary carcinoma, two chronic pancreatitis, and one renal cell carcinoma), and hilar biliary obstruction (intrahepatic cholangiocarcinoma) in two patients. The reasons for failed cannulation were as follows: long narrow distal segment of the bile duct in seven, unidentifiable papilla in four, tumor invasion of the papilla in three, and intradiverticular papilla in two patients. Nine patients underwent EUS-RV in the same session as the prior failed ERCP. In the other seven patients, EUS-RV was performed in a different session because written informed consent had not been obtained prior to ERCP (Table 2).

### 3.2. Approach Routes and Needle Sizes According to the Biliary Disorders

Of 16 patients, the DEHBD/D2 route was attempted in 10 (62.5%) patients, and successful biliary cannulation was achieved in all of these patients. Of the six patients unfit for the DEHBD/D2 route, the PEHBD/D1 route was attempted in five (31.3%) patients and successful biliary cannulation was achieved in four of these patients. One patient with failed biliary puncture and another patient unfit for the PEHBD/D1 route was moved to IHBD approaches. Of these patients, the LIHBD/S route was used in one patient with successful cannulation and another patient was moved to the RIHBD/D1 route because of tumor invasion to the left hepatic lobe. Consequently, biliary cannulation finally succeeded. A 19 gauge needle was used in all patients but one, for whom a 22 gauge needle was used. Successful cannulation was achieved in all patients eventually (Figure 7A).

In all patients with bile duct stones (*n* = 5), the DEHBD/D2 route was applied and successful biliary cannulation was achieved (Figure 7B). In patients with distal biliary obstructions (*n* = 9), the DEHBD/D2 route was attempted in four patients, resulting in successful biliary cannulation. The reasons for not applying the DEHBD/D2 route included scope instability in three, tumor invasion of the DEHBD in one, and an active ulcer at the D2 in one patient. Of these five patients, four patients underwent EUS-RV with the PEHBD/D1 route and succeeded in biliary cannulation. The one remaining patient was moved to the LIHBD/S route because of a pancreatic cyst on the puncture line of the PEHBD/D1 route, and successful cannulation was obtained (Figure 7C). In patients with hilar biliary obstructions (*n* = 2), the DEHBD/D2 route was attempted in one patient with a 22 gauge needle, resulting in successful cannulation. In another patient, the DEHBD/D2 route was not used because of the non-dilated DEHBD (1 mm), and the PEHBD/D1 route was attempted with a 19 gauge needle. However, the puncture of the PEHBD failed due to inadequate dilatation of the PEHBD (3 mm). Although the LIHBD/S route should be the next step, the RIHBD/D1 route was used because of tumor invasion to the left hepatic lobe, and successful cannulation was achieved (Figure 7D).

### 3.3. Procedural Details According to Approach Routes

Comparing the four approach routes, the diameter of the punctured duct, the number of punctures, the size of the needle, the type of the echoendoscope, the use of a catheter for guidewire manipulation, the cannulation methods, and the biliary drainage methods were not significantly different (Table 3). A 19 gauge needle was used in all patients, except for one patient with a hilar biliary obstruction due to intrahepatic cholangiocarcinoma. In this patient, the DEHBD was so thin (2 mm) that a 22 gauge needle was used in the DEHBD/D2 route, and the puncture succeeded in one attempt. Biliary drainage tubes were placed in all patients, except for one patient with bile duct stones in whom no ERCP catheter was used during the guidewire manipulation. Although the time from the puncture to the guidewire placement was not significantly different between the four groups, the time in the DEHBD/D2 route was significantly shorter than that in the other routes (*p* = 0.014, Figure 8A).

### 3.4. Outcomes and Adverse Events of EUS-RV According to Approach Routes

The biliary puncture, guidewire placement, and cannulation succeeded in all EUS-RV attempts, except for one attempt with the PEHBD/D1 route in which the biliary puncture failed due to non-dilated PEHBD, as described previously (Table 4). The time from the echoendoscope insertion to cannulation in the DEHBD/D2 route was shorter in comparison with the other routes, although there was no significant difference (*p* = 0.07, Figure 8B).

Adverse events occurred in one (6.3%) patient with moderate pancreatitis after EUS-RV with the PEHBD/D1 route and the along-the-wire cannulation method. There were no adverse events related to the bile leak or hemorrhage.

### 3.5. Cannulation Methods

In the cannulation attempts, the over-the-wire method, along-the-wire method, and hitch-and-ride methods were used in four, five, and seven patients, respectively (Table 5). Conversion to another method was required in one patient in each method. When using the over-the-wire method, the guidewire came off the snare in the scope channel in one patient, and the along-the-wire method was alternatively used. When using the along-the-wire method, the guidewire could be inserted into only the pancreatic duct during a WGC attempt in one patient, and TPPS was performed. When using the hitch-and-ride method, the slit of the catheter tip failed to snap the EUS-placed guidewire in one patient, so that WGC with a normal catheter was performed along the EUS-placed guidewire.

The time from the duodenoscope insertion to cannulation was shorter in the hitch-and-ride method (7 min) than that in the over-the-wire and along-the-wire methods (11 and 14 min, respectively), although there was no significant difference between the three cannulation methods (p = 0.206).

## 4. Discussion

In the current study, EUS-RV performed in accordance with our method succeeded in all patients. The first attempted methods allowed successful biliary cannulation in all except for one (6.3%) patient with failed biliary puncture, while there were no cases with failed guidewire manipulation. Adverse events occurred in 6.3% (one of 16) of the patients with pancreatitis, whereas any adverse events related to the bile leak did not occur. The hitch-and-ride method could not show a definite predominance over the other methods in the time and success rate of biliary cannulation.

Since the biliary puncture requires a short time, has a small risk of bile leak even after failure, and is the first step of the EUS-RV procedure, changing the approach route after failed biliary puncture is not so serious. Alternately, the guidewire manipulation is the most challenging and time-consuming step [14] and has a risk of bile leak when biliary drainage fails. Then, our method was made to focus on guidewire manipulation on the basis of a previously proposed algorithm reported by Iwashita et al. [3], in which the DEHBD/D2 route was the first-line approach, followed by both the PEHBD/D1 and LIHBD/S routes as the second-line approaches. In our method, we modified their algorithm in some ways. First, the RHIBD/D1 route was added to improve the adaptability of EUS-RV for various biliary disorders. Second, the PEHBD/D1 route was prioritized over IHBD routes because of the shorter distance between the puncture site and the ampulla and the less necessity of passing through the strictures. Third, a 22 gauge needle for an inadequately dilated bile duct in the DEHBD/D2 route was included. Fourth, insertion of a catheter was proposed in cases with poor guidewire manipulation.

Consequently, the DEHBD/D2 route was applied to 62.5% (10/16) of the patients and allowed successful cannulation in all the patients, requiring a shorter time for guidewire manipulation than the other routes. In addition, the whole time for EUS-RV with the DEHB/D2 route was also shorter than for the other routes, although there was no significant difference, possibly due to other factors, including the cannulation time. Although multiple attempts at the biliary puncture were needed in 40% (four of 10) of patients due to a thin DEHBD, there were no significant differences in the puncture number between approach routes. The PEHBD/D1 route was applied to 31.3% (five of 16) of the patients, and the success rate was 80% (4/5). The cause of unsuccess was failed biliary puncture due to a thin PEHBD (3 mm). The PEHBD lacked fixing by surrounding solid organs, unlike the IHBD or DEHBD, and needle handling in the long scope position was poor so that the puncture of the PEHBD might have been more difficult than that of the IHBD or DEHBD, when enough dilation of the bile duct was absent. On the contrary, the non-dilated DEHBD (2 mm) in a hilar biliary obstruction case was successfully punctured in one attempt, although a 22 gauge needle was used, possibly owing to fixing by the pancreas and favorable needle handling in the short scope position. We did not use 22 gauge needles except for in DEHBD/D2 route because the manipulation of a 0.018 inch guidewire was worse than that of a 0.025 inch guidewire. Indeed, Martínez et al. reported that the success rate of the manipulation of a 0.018 inch guidewire through a 22 gauge needle was only 80.6% (25 of 31 cases) [19]. Some authors mentioned a method for cases with thin bile ducts in which the bile duct was punctured with a 22 gauge needle and a contrast was injected to dilate the bile duct, followed by re-puncture with a 19 gauge needle [4,20]. However, we did not use this method because of the additional cost of the needle and the potential risk of a bile leak. Of four patients with successful punctures in the PEHBD/D1 route, the guidewire placement was successful in all patients, although a longer time was required than for the DEHBD/D2 route. Some authors mentioned that advancing the guidewire to the duodenum via the PEHBD/D1 route was troublesome and often failed owing to the needle direction toward the hepatic hilum [3,5]. To overcome this problem, we preferably used EG-580UT over GF-UCT260. The high flexibility scope shaft and the wide ranges of the angle and elevator of EG-580UT could allow the needle to advance toward the distal side of the bile duct (Figure 2b). When the needle direction was not favorable for advancing the guidewire during the puncture, we changed the needle direction to the caudal side by pushing the scope to the cranial side and by using the up-angle of the scope (Figure 9). If advancing the guidewire toward the ampulla was impossible, even when using maneuvers of the scope, we inserted a catheter into the bile duct to assist the guidewire manipulation [5,21]. IHBD approaches were needed in only two patients. The LIHBD/S route was used in the patient with a distal biliary obstruction and a pancreatic head cyst, and the guidewire passage through the stricture required a very long time (19 min), even when using the catheter support, because of the long, winding, and wide bile duct between the puncture point and the stricture (Figure 3B). The RIHBD/D1 route was applied in one case with the difficult puncture of the EHBD or LIHBD.

Discussing according to the biliary disorders, the DEHBD/D2 route was successfully performed in 100% (five of five) of cases with bile duct stones. Generally, the puncture of the DEHBD from the D2 in cases with bile duct stones was considered to be difficult because of inadequate intraductal pressure [3]. To prevent the collapse of the bile duct by the tip of the needle, we pushed the needle handle as quickly as possible, hitting the needle stopper with the maximum force as needed. With regard to distal biliary obstruction, the DEHBD/D2 route was used in only 44.3% (four of nine) of the patients mainly because of the unstable scope position due to the malformation of the duodenum caused by tumor infiltration or chronic pancreatitis. The PEHBD/D1 route was successfully used in all remaining patients but one for whom the LIHBD/S route was used because of an intervening pancreatic cyst. Since the guidewire passage through the distal biliary stricture in the LIHBD/S route was extremely difficult because of the long, winding, and wide bile duct in this case and in previous reports [7,21,22], the PEHBD/D1 route might be better, especially in distal biliary obstruction, than IHBD approaches when the DEHBD/D2 route is impossible or failed. As for hilar biliary obstruction, the puncture of the EHBD was exceedingly challenging because of its extreme thinness due to a lack of the influx of bile juice, so the DEHBD/D2 route was successfully used in only 50% (one of two) of patients, even when using a 22 gauge needle, whereas the RIHBD/D1 route was needed in another patient. The RIHBD/D1 route was a useful option for hilar biliary obstruction when the LIHBD/S route was unavailable due to the tumor location.

The number of patients in the present study was so small that the validity of our method could not be confirmed. To complement the small number of patients, we performed a review of the literature that described the rates of success and adverse events for each approach route separately and had five patients or more (Table 6). If the distinction between DEHBD/D2 and PEHBD/D1 was uncertain in cases with the EHBD approach, those cases were excluded from the review. According to this review, the DEHBD/D2 and PEHBD/D1 routes were equal in terms of the success rate (92.1% vs. 88.3%, *p* = 0.477) and the adverse event rate (10.2% vs. 9.3%, *p* = 1). As for the comparison of the DEPBD/D2 and PEHBD/D1 routes, Iwashita et al. directly compared them in a prospective study, and the higher cannulation success rate of the DEHBD/D2 route than that of the PEHBD/D1 route was demonstrated [3]. In addition, the shorter guidewire manipulation time in the DEHBD/D2 route was showed in the current study. Given these facts, the DEHBD/D2 route was likely to be suitable for the first-line approach. In the review, the PEHBD/D1 route was better than the LIHBD/S route in terms of success rate (88.3% vs. 78.8%, *p* = 0.059) and adverse event rate (9.3% vs. 16.9%, *p* = 0.119), although there were no significant differences. Moreover, Dhir et al. reported that the time between the introduction of the echoendoscope and the introduction of the duodenoscope in the PEHBD/D1 route was significantly shorter than that in the LIHBD/S route [22]. Consequently, our strategy in which the PEHBD/D1 route should take precedence over the LIHBD/S route was considered to be reasonable. As for the RIHBD/D1 route, we could not validate our strategy because there was only one case report including the RIHBD/D1 route [23].

Considering both the literature review and our experiences, we proposed a new algorithm for EUS-RV based on our method (Figure 10). The DEHBD/D2 route with a 19 gauge needle is the first-line method. If the puncture is impossible or fails due to inadequate EHBD dilation, the DEHBD/D2 route with a 22 gauge needle is recommended. If the puncture is impossible or fails even when using a 22 gauge needle or due to scope instability or tumor invasion, the PEHBD/D1 route, LIHBD/S route, and RIHBD/D1 routes are recommended in this order with a 19 gauge needle. When the guidewire is hard to advance to the duodenum, catheter insertion into the bile duct is suggested.

We previously reported that the hitch-and-ride method could shorten the cannulation time [16]. In the present study, the hitch-and-ride method had a tendency to shorten the cannulation time compared with the other methods. However, there was no statistically significant difference. The reasons for the negative outcome in the present study were considered to be the small number of patients and the heterogeneity of the hand-made catheters. Despite the lack of decisive superiority, we recommend the hitch-and-ride method because of the possibility of reducing the cannulation time and because of its easiness.

Adverse events included only one instance of pancreatitis (6.3%) in the present study. Although some authors explained that EHBD approaches have a larger risk of bile leak than IHBD approaches due to a lack of the tamponade effect in the surrounding liver parenchyma [2,7,11,12], we did not encounter any adverse events related bile leak, notwithstanding the use of EHBD routes in 87.5% (14 of 16) of patients. Dhir et al. reported that a higher incidence of adverse events was seen with the LIHBD/S route than with the PEHBD/D1 route and suggested that the reason might be the dynamic movement of the left lobe of the liver with respiration [22]. Actually, the adverse event rate was higher in the LIHBD/S route than in EHBD approaches in our literature review. We believe that a major cause of bile leak is failed cannulation after successful biliary puncture, as well as a long procedure time between the puncture and the biliary drainage. Thus, our strategy, which highlights the guidewire manipulation over the biliary puncture, could prevent bile leak. Moreover, we deployed biliary stents or naso-biliary tubes across the puncture site where possible to prevent bile leak, particularly when a catheter was inserted via the puncture point.

There are some limitations in the present study. First, the retrospective design including a small number of patients was the major limitation of this study. Second, all procedures were performed at a single center and by a single endoscopist. Third, the timing of the EUS-RV was not definite, which was likely to affect the rates of successful cannulation and adverse events.

In conclusion, the new proposed algorithm, which highlights guidewire manipulation, might facilitate EUS-RV for failed biliary cannulation. The DEHBD/D2 route was the first-line approach but was sometimes unavailable, especially in cases of distal and hilar biliary obstruction, because of scope instability and the thin bile duct, respectively. The PEHBD/D1 route was the second-line approach. Particularly in cases of distal biliary obstruction, the PEHBD/D1 route might be more suitable than the IHBD approach because guidewire manipulation in the IHBD approach was extremely difficult. In cases of hilar biliary obstruction, the biliary puncture in the PEHBD/D1 route might be difficult, and the puncture of the dilated IHBD was useful. If the LIHBD/S route was unavailable due to the tumor location, the RIHBD/D1 route was recommended. Further investigation in a prospective study with a large cohort is warranted.

## Figures and Tables

**Figure 1 jcm-09-03879-f001:**
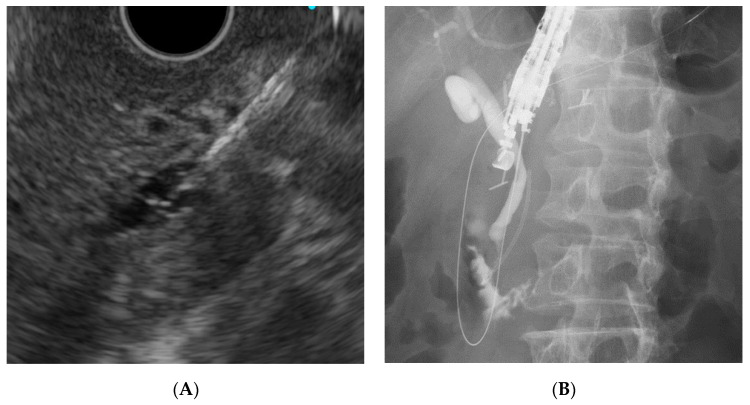
The distal extrahepatic bile duct/duodenal second part (DEHBD/D2) route. Endoscopic ultrasound image (**A**) showing needle puncture of the distal extrahepatic bile duct from the duodenal second part. Fluoroscopic image (**B**) showing that the guidewire passed through the papilla into the duodenum.

**Figure 2 jcm-09-03879-f002:**
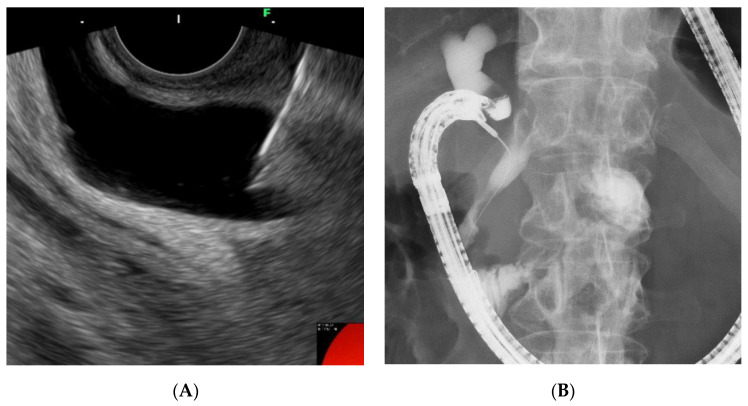
The proximal extrahepatic bile duct/duodenal bulb (PEHBD/D1) route. Endoscopic ultrasound image (**A**) showing needle puncture of the proximal extrahepatic bile duct from the duodenal bulb. Fluoroscopic image (**B**) showing that the guidewire passed through the stricture and papilla into the duodenum.

**Figure 3 jcm-09-03879-f003:**
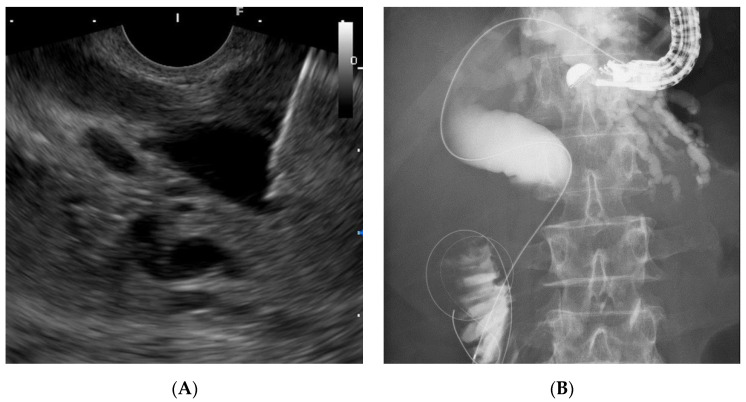
The left intrahepatic bile duct/stomach (LIHBD/S) route. Endoscopic ultrasound image (**A**) showing needle puncture of the left intrahepatic bile duct (B3) from the stomach. Fluoroscopic image (**B**) showing that the guidewire passed through the stricture and papilla into the duodenum.

**Figure 4 jcm-09-03879-f004:**
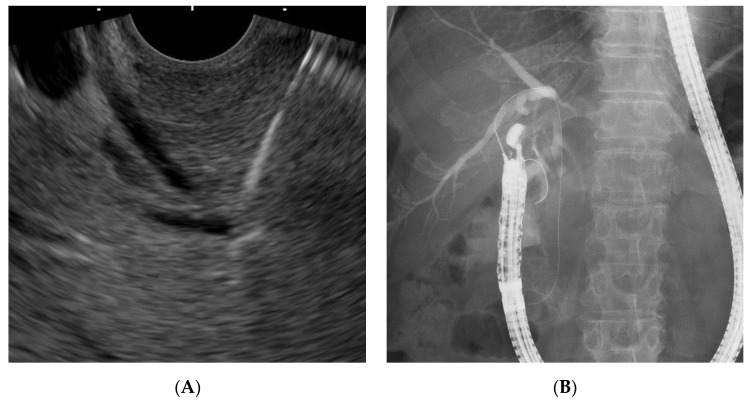
The right intrahepatic bile duct/duodenal bulb (RIHBD/D1) route. Endoscopic ultrasound image (**A**) showing needle puncture of the right intrahepatic bile duct (B6) from the duodenal bulb. Fluoroscopic image (**B**) showing that the guidewire passed through the stricture and papilla into the duodenum.

**Figure 5 jcm-09-03879-f005:**
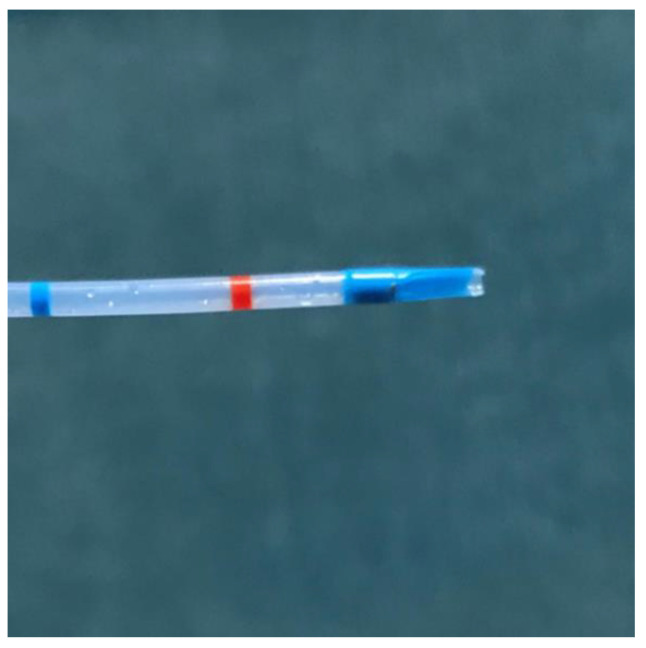
A hand-made “hitch-and-ride” catheter with a slit at its distal end.

**Figure 6 jcm-09-03879-f006:**
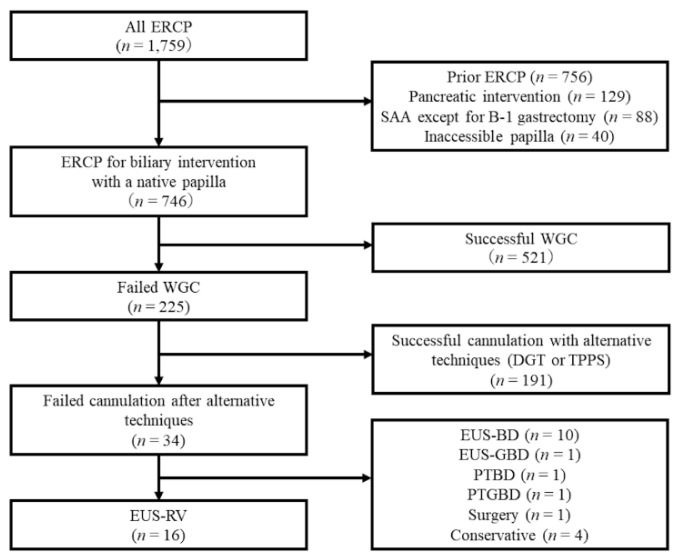
Flowchart of the study patients. ERCP: endoscopic retrograde cholagiopancreatography; SAA: surgically altered anatomy; WGC: wire-guided cannulation; DGT: double guidewire technique; TPPS: transpancreatic precut sphincterotomy; EUS-RV: endoscopic ultrasound-guided rendezvous; EUS-BD: endoscopic ultrasound-guided biliary drainage; PTBD: percutaneous transhepatic biliary drainage; EUS-GBD: endoscopic ultrasound-guided gallbladder drainage; PTGBD: percutaneous transhepatic gallbladder drainage

**Figure 7 jcm-09-03879-f007:**
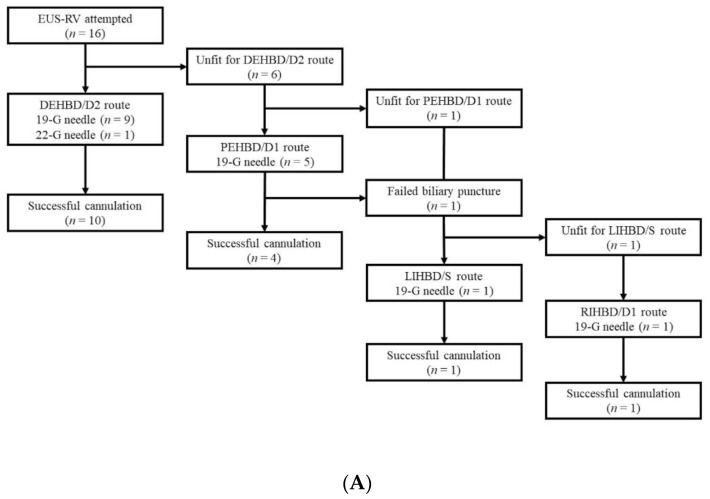
Approach routes and needle sizes according to the biliary disorders. All patients (**A**), bile duct stones (**B**), distal biliary obstruction (**C**), and hilar biliary obstruction (**D**). DEHBD/D2: distal extrahepatic bile duct/duodenal second part; PEHBD/D1: proximal extrahepatic bile duct/duodenal bulb; LIHBD/S: left intrahepatic bile duct/stomach; RIHBD/D1: right intrahepatic bile duct/duodenal bulb; EUS-RV: endoscopic ultrasound-guided rendezvous.

**Figure 8 jcm-09-03879-f008:**
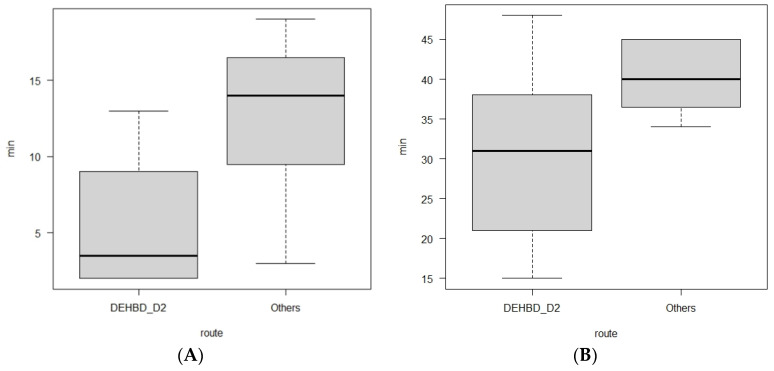
Box plots showing comparisons of procedural time between the DEBHD/D2 route and the others. The time from the puncture to the guidewire placement in the DEHBD/D2 route is significantly shorter than that in the others (*p* = 0.014) (**A**). The time from echoendoscope insertion to cannulation in the DEHBD/D2 route is shorter than that in the others but not significant (*p* = 0.07) (**B**). Bold horizontal lines inside boxes indicate median. Upper and lower sides of boxes indicate upper quartile and lower quantile, respectively. Short lines above and below boxes indicate maximum and minimum, respectively. DEHBD/D2: distal extrahepatic bile duct/duodenal second part.

**Figure 9 jcm-09-03879-f009:**
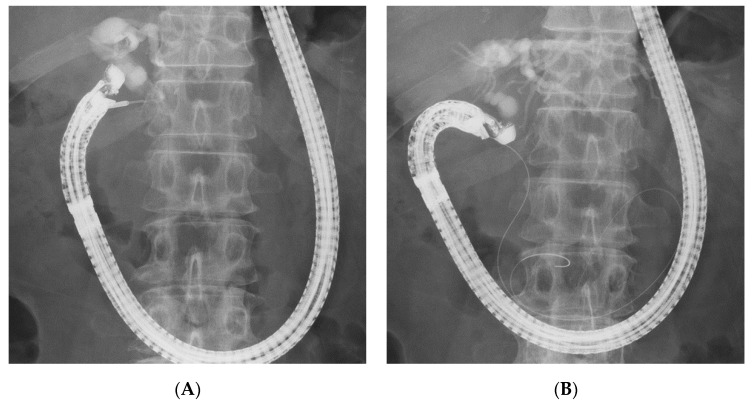
Changing the needle direction after biliary puncture in the PEHBD/D1 route. The proximal extrahepatic bile duct was punctured from the duodenal bulb with an unfavorable needle direction (**A**). Changing the needle direction to the caudal side by pushing the scope to the cranial side and using the up-angle of the scope (**B**). PEHBD/D1, proximal extrahepatic bile duct/duodenal bulb.

**Figure 10 jcm-09-03879-f010:**
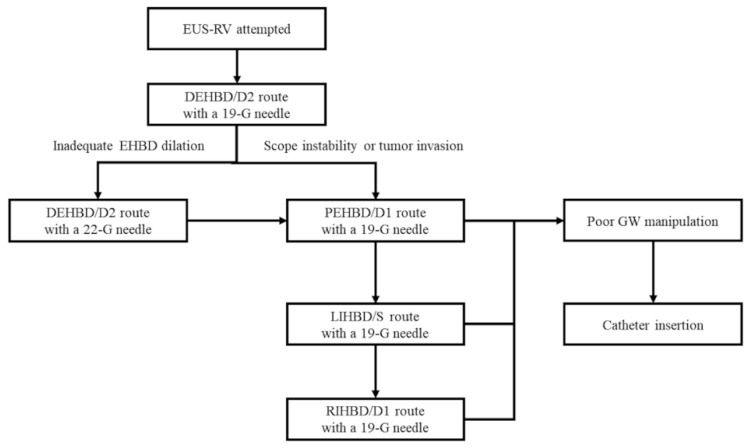
A proposed new algorithm for endoscopic ultrasound-guided rendezvous (EUS-RV). DEHBD/D2: distal extrahepatic bile duct/duodenal second part; PEHBD/D1: proximal extrahepatic bile duct/duodenal bulb; LIHBD/S: left intrahepatic bile duct/stomach; RIHBD/D1: right intrahepatic bile duct/duodenal bulb; GW: guidewire.

**Table 1 jcm-09-03879-t001:** Comparison of the approach routes.

	DEHBD/D2 Route	PEHBD/D1 Route	LIHBD/S Route	RIHBD/D1 Route
Scheme	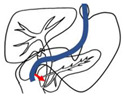	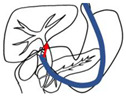	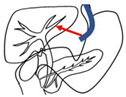	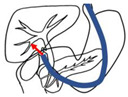
Scope position	Short	Long	Short	Long
Scope stability	Poor	Good	Mediocre	Good
Needle handling	Good	Poor	Good	Poor
Bile duct diameter *	Mediocre	Good	Poor	Poor
GW manipulation	Good	Mediocre	Mediocre	Poor

* Affected by biliary disorders and the site of biliary obstruction. DEHBD/D2: distal extrahepatic bile duct/duodenal second part; PEHBD/D1: proximal extrahepatic bile duct/duodenal bulb; LIHBD/S: left intrahepatic bile duct/stomach; RIHBD/D1: right intrahepatic bile duct/duodenal bulb; GW: guidewire.

**Table 2 jcm-09-03879-t002:** Baseline characteristics.

Age, median (range), y	74 (40–89)
Sex, male/female, *n*	13/3
Indication, *n* (%)	
Bile duct stones	5 (31.3)
Distal biliary obstruction	9 (56.3)
Pancreatic carcinoma	3 (18.8)
Ampullary carcinoma	3 (18.8)
Chronic pancreatitis	2 (12.5)
Renal cell carcinoma	1 (6.3)
Hilar biliary obstruction	2 (12.5)
Intrahepatic cholangiocarcinoma	2 (12.5)
Reason for failed cannulation, *n* (%)	
Long NDS	7 (43.8)
Unidentifiable papilla	4 (25.0)
Tumor invasion of the papilla	3 (18.8)
Intradiverticular papilla	2 (12.5)
Time for EUS-RV after failed ERCP, *n* (%)	
Same session	9 (56.3)
Different session	7 (43.8)

NDS: narrow distal segment; EUS-RV: endoscopic ultrasound-guided rendezvous; ERCP: endoscopic retrograde cholangiopancreatography.

**Table 3 jcm-09-03879-t003:** Details of the EUS-RV procedures.

	DEHBD/D2 Route	PEHBD/D1 Route	LIHBD/S Route	RIHBD/D1 Route	*p*
	(*n* = 10)	(*n* = 5)	(*n* = 1)	(*n* = 1) *	
Indication, *n* (%)					0.123
Bile duct stones	5 (50)	0	0	0	
Distal biliary obstruction	4 (40)	4 (80)	1 (100)	0	
Hilar biliary obstruction	1 (10)	1 (20)	0	1 (100)	
Diameter of a punctured duct, median (range), mm	7.5 (2–22)	12 (3–20)	13	4	0.593
Number of punctures, median (range), *n*	1 (1–3)	1 (1)	1	2	0.22
Size of the needle, *n* (%)					1
19G	9 (90)	5 (100)	1 (100)	1 (100)	
22G	1 (10)	0	0	0	
Echoendoscope, *n* (%)					0.37
EG-580UT	6 (60)	5 (100)	1 (100)	1 (100)	
GF-UCT260	4 (40)	0	0	0	
Catheter use in GW manipulation, *n* (%)					0.143
Yes	1 (10)	2 (40)	1 (100)	0	
No	9 (90)	2 (40)	0	1 (100)	
Cannulation method, *n* (%)					0.86
Over-the-wire	2 (20)	1 (20)	1 (100)	0	
Along-the-wire	4 (40)	1 (20)	0	0	
Hitch-and-ride	4 (40)	2 (40)	0	1 (100)	
Biliary drainage, *n* (%)					0.27
None	1 (10)	0	0	0	
Plastic stent	4 (40)	3 (60)	0	0	
Naso-biliary tube	4 (40)	0	0	1 (100)	
Self-expandable metal stent	1 (10)	1 (20)	1 (100)	0	
Time from the puncture to GW placement, median (range), min	3.5 (2–13)	14 (3–19)	19	6	0.063

* Conversion from the PEHBD/D1 route. EUS-RV: endoscopic ultrasound-guided rendezvous; DEHBD/D2: distal extrahepatic bile duct/duodenal second part; PEHBD/D1: proximal extrahepatic bile duct/duodenal bulb; LIHBD/S: left intrahepatic bile duct/stomach; RIHBD/D1: right intrahepatic bile duct/duodenal bulb; GW: guidewire.

**Table 4 jcm-09-03879-t004:** Outcomes of EUS-RV.

	DEHBD/D2 Route	PEHBD/D1 Route	LIHBD/S Route	RIHBD/D1 Route	*p*
	(*n* = 10)	(*n* = 5)	(*n* = 1)	(*n* = 1) *	
Successful biliary puncture, *n* (%)	10 (100)	4 (80)	1 (100)	1 (100)	0.412
Successful GW placement, *n* (%)	10 (100)	4 (80)	1 (100)	1 (100)	0.412
Successful cannulation, *n* (%)	10 (100)	4 (80)	1 (100)	1 (100)	0.412
Conversion to the other route, *n* (%)	0	1 (20)	0	0	0.412
Final biliary cannulation, *n* (%)	10 (100)	5 (100)	1 (100)	-	1
Time from echoendoscope insertion to cannulation (range), min	31 (15–48)	37 (34–45)	45	45	0.199
Adverse events, *n* (%)	0	1 (20)	0	0	0.412
Adverse events	-	moderate pancreatitis	-	-	

* Conversion from the PEHBD/D1 route. EUS-RV: endoscopic ultrasound-guided rendezvous; DEHBD/D2: distal extrahepatic bile duct/duodenal second part; PEHBD/D1: proximal extrahepatic bile duct/duodenal bulb; LIHBD/S: left intrahepatic bile duct/stomach; RIHBD/D1: right intrahepatic bile duct/duodenal bulb; GW: guidewire.

**Table 5 jcm-09-03879-t005:** Comparison of cannulation methods.

	Over-the-Wire	Along-the-Wire	Hitch-and-Ride	*p*
	(*n* = 4)	(*n* = 5)	(*n* = 7)	
Conversion to the other method, *n* (%)	1 (25)	1 (20)	1 (14)	1
Reason for conversion	GW loss	Difficult cannulation	Failed slit snapping at GW	
Method of conversion	Along-the-wire	TPPS	Along-the-wire	
Final biliary cannulation, *n* (%)	4 (100)	5 (100)	7 (100)	1
Time from scope exchange to cannulation, median (range), min	14 (13–17)	11 (3–26)	7 (4–14)	0.206
Adverse events, *n* (%)	0	1 (20)	0	0.562
Adverse events	-	moderate pancreatitis	-	

GW: guidewire; TPPS: transpancreatic precut sphincterotomy.

**Table 6 jcm-09-03879-t006:** Review of published studies with ≥5 patients.

		Success, *n* (%)									Adverse Events, *n* (%)								
**Authors**	**Year**	**DEHBD/D2**			**PEHBD/D1**			**LIHBD/S**			**DEHBD/D2**			**PEHBD/D1**			**LIHBD/S**		
Kim et al. [9]	2010	12/15 (80)									1/15 (6.7)								
Dhir et al. [10]	2012				57/58 (98.3)									2/58 (3.4)					
Iwashita et al. [14]	2012							4/9 (44.4)									1/9 (11.1)		
Dhir et al. [22]	2013				18/18 (100)			16/17 (94.1)						1/18 (5.6)			7/17 (41.1)		
Kawakubo et al. [7]	2013	5/5 (100)			4/4 (100)			5/5 (100)			1/5 (20)			1/4 (25)			0/5 (0)		
Park et al. [12]	2013							3/6 (50)									0/6 (0)		
Poincloux et al. [24]	2015							5/5 (100)									0/5 (0)		
Bill et al. [25]	2016				19/25 (76)									4/25 (16)					
Iwashita et al. [3]	2016	10/10 (100)			3/5 (60)			3/4 (75)			1/10 (10)			0/5 (0)			2/4 (50)		
Tang et al. [26]	2016				20/24 (83.3)			0/1 (0)						N/A			N/A		
Okuno et al. [27]	2017	3/4 (75)			4/7 (57.1)			12/16 (75)			2/4 (50)			1/7 (14.3)			1/16 (6.2)		
Iwashita et al. [21]	2018	14/14 (100)			1/1 (100)			11/13 (84.6)			N/A			N/A			N/A		
Shiomi et al. [5]	2018	4/5 (80)			6/7 (85.7)			7/8 (87.5)			0/5 (0)			2/7 (28.6)			1/8 (12.5)		
Present study		10/10 (100)			4/5 (80)			1/1 (100)			0/10 (0)			1/5 (20)			0/1 (0)		
Total		58/63 (92.1)			136/154 (88.3)			67/85 (78.8)			5/49 (10.2)			12/129 (9.3)			12/71 (16.9)		

				*p* = 0.477			*p* = 0.059						*p* = 1			*p* = 0.119			
		*p* = 0.038										*p* = 0.426							

DEHBD/D2: distal extrahepatic bile duct/duodenal second part; PEHBD/D1: proximal extrahepatic bile duct/duodenal bulb; LIHBD/S: left intrahepatic bile duct/stomach; RIHBD/D1: right intrahepatic bile duct/duodenal bulb; N/A: not applicable.

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
