# Peer review of "A Proposed Algorithm for Endoscopic Ultrasound-Guided Rendezvous Technique in Failed Biliary Cannulation"

_jcm, 2020, doi:10.3390/jcm9123879_

Round 1

Reviewer 1 Report

The author proposed a novel algorithm for EUS-RV in failed biliary cannulation that focused on the importance of guidewire manipulation to perform safe and reliable EUS-RV. This study is well conducted and the data is presented clearly. And also this paper has a thoughtful discussion. Furthermore, these findings will be of interest to endoscopist in the field. There are a few comments as follows.

<Major comment>

# 1

A very logical and well thought out algorithm has been proposed. However, as pointed out in the limitation section, the main problem with this study is that the number of cases is very small. In particular, the fact that the LIHBD/S route or the RIHBD/D1 route is one case, which is a lack of evidence to suggest that the PEHBD/D1 route is 2nd step in the proposed algorithm. Is it possible that the superiority of the LIHBD/S or RIHBD/D1 route over the PEHBD/D1 route may be demonstrated when considering a large number of cases? To date, there are many published data on EUS-RV. The authors should present the success rates and adverse event rates for the PEHBD/D1, LIHBD/S, and RIHBD/D1 routes from previous reports as a literature review. Complementing the small number of cases in this study can show the validity of the algorithm.

# 2

In Figure 9, the PEHBD/D1 route, LIHBD/S route, and RIHBD/D1 route all have only 19-G needles. In clinical practice, if the bile ducts are thin and sometimes difficult to puncture with a 19-G needle, a 22-G needle should be used. A 2-step procedure, in which the thin bile ducts are first dilated by cholangiography with a 22-G needle and then re-punctured with a 19-G needle, has also been reported. The algorithm should be restructured including these points.

Author Response

Response to Reviewer 1 Comments

Point 1

A very logical and well thought out algorithm has been proposed. However, as pointed out in the limitation section, the main problem with this study is that the number of cases is very small. In particular, the fact that the LIHBD/S route or the RIHBD/D1 route is one case, which is a lack of evidence to suggest that the PEHBD/D1 route is 2nd step in the proposed algorithm. Is it possible that the superiority of the LIHBD/S or RIHBD/D1 route over the PEHBD/D1 route may be demonstrated when considering a large number of cases? To date, there are many published data on EUS-RV. The authors should present the success rates and adverse event rates for the PEHBD/D1, LIHBD/S, and RIHBD/D1 routes from previous reports as a literature review. Complementing the small number of cases in this study can show the validity of the algorithm.

Response 1

Thank you for your great comment. We performed the literature review as your request to complement the small number of cases in the present study, particularly in IHBD approach. Consequently, literature review revealed the superiority of PEHBD/D1 route over LIHBD/S route in terms of success rate and adverse events rate. We added Table 6 and text in the discussion section, line 405-423. As for RIHBD/D1 route, only one case has been reported.

Point 2

In Figure 9, the PEHBD/D1 route, LIHBD/S route, and RIHBD/D1 route all have only 19-G needles. In clinical practice, if the bile ducts are thin and sometimes difficult to puncture with a 19-G needle, a 22-G needle should be used. A 2-step procedure, in which the thin bile ducts are first dilated by cholangiography with a 22-G needle and then re-punctured with a 19-G needle, has also been reported. The algorithm should be restructured including these points.

Response 2

Thank you for your precise comment. We did not use the 2-step procedure, because we concerned about the risk of the bile leak and increased cost with 2 needles. I add the text in the discussion section, line 361-364.

Reviewer 2 Report

I read with interest the manuscript by Saburo Matsubara and colleagues, about the EUS-rendezvous technique for biliary drainage in failed ERCP cannulation.

This is a very relevant topic, with several trials, reviews and meta-analysis published in the last years.

The manuscript introduction is very detailed as well the patients and methods section, although I would suggest to include the adverse events classification in the outcome section, instead in the statistical part.

However, it is in the results and discussion sections, where I found the most critical points.

In fact, Authors cannot assess the appropriateness of the proposed algorithm, since this is a retrospective analysis.

So the conclusion of the manuscript cannot be accepted.

Authors should deeply reconsider the structure of their study.

Author Response

Response to Reviewer 2 Comments

Point 1

I would suggest to include the adverse events classification in the outcome section, instead in the statistical part.

Response 1

Thank you for your precise comment. We revised as your request. The description about adverse events classification was moved to “2.6. Outcome measures” section from “2.7. Statistical analyses” section.

Point 2

In fact, Authors cannot assess the appropriateness of the proposed algorithm, since this is a retrospective analysis. So the conclusion of the manuscript cannot be accepted. Authors should deeply reconsider the structure of their study.

Response 2

Thank you for your great comment. We thoroughly agree with your opinion.

We performed EUS-RV in accordance with the prearranged method. In this study, we proposed an algorithm based on the analysis of the prearranged method. To complement the small number of cases in the present study, we performed the literature review for validation. We added the result of the review as Table 6 and text in the discussion section, line 405-423.

As a result, our proposed algorithm did not conflict with the literature review. However, definitive conclusion cannot be obtained because this was a retrospective study. Therefore, I changed my conclusion in the manuscript and in the abstract, to nonconclusive expression.  

Reviewer 3 Report

I have read your article with great interest. It's a great article to try to standardize  a difficult technique like the EUS-guided rendezvous.

It is a pity that the small number of patients included (16) does not allow us to draw solid conclusions in this regard.

There are several issues:
In figure 6, the numbers do not match. 746 - 531 = 215 failed WGC. I think the correct number must be 521 Succesful WGC.

In figure 6, Other = 7, What were these others?

What were the reasons why the rendezvous was not performed in the same session as the ERCP?

Author Response

Response to Reviewer 3 Comments

Point 1

In figure 6, the numbers do not match. 746 - 531 = 215 failed WGC. I think the correct number must be 521 Succesful WGC.

Response 1

Thank you for your comment. We revised figure 6 as your indication. Number of successful WGC was changed from 531 to 521.

Point 2

In figure 6, Other = 7, What were these others?

Response 2

Thank you for your comment. Seven cases included EUS-GBD (n = 1), PTGBD (n = 1), surgery (n = 1), and conservative treatment ( n = 4). We revised figure 6.

Point 3

What were the reasons why the rendezvous was not performed in the same session as the ERCP?

Response 3

Thank you for comment. In seven patients, EUS-RV was performed in the different session because written informed consent had not been obtained prior to ERCP. We added this point in the result section, line 207-208.

Round 2

Reviewer 1 Report

The authors answered all the questions.

Reviewer 2 Report

Dear Editor,

       I read with interest the revised version of the manuscript.

Although the study is still not clearly relevant in term of scientific novelties, I think that the Authors, accepting all the comments,  could give a new perspective to the results.

The proposed version of the manuscript is a captivating analysis of EUS-RV, resulting in a stimulating algorithm.